# Regorafenib-Attenuated, Bleomycin-Induced Pulmonary Fibrosis by Inhibiting the TGF-β1 Signaling Pathway

**DOI:** 10.3390/ijms22041985

**Published:** 2021-02-17

**Authors:** Xiaohe Li, Ling Ma, Kai Huang, Yuli Wei, Shida Long, Qinyi Liu, Deqiang Zhang, Shuyang Wu, Wenrui Wang, Guang Yang, Honggang Zhou, Cheng Yang

**Affiliations:** 1State Key Laboratory of Medicinal Chemical Biology, College of Pharmacy and Tianjin Key Laboratory of Molecular Drug Research, Nankai University, Tianjin 300353, China; lixiaohe908@163.com (X.L.); mal930602@163.com (L.M.); kaihuang0819@163.com (K.H.); weiyuli2021@126.com (Y.W.); starlongsd@163.com (S.L.); jiukn365@126.com (Q.L.); 13345278851@163.com (D.Z.); shuyang_wu@163.com (S.W.); wwr15641208039@163.com (W.W.); 2Tianjin Key Laboratory of Molecular Drug Research, Tianjin International Joint Academy of Biomedicine, Tianjin 300457, China

**Keywords:** pulmonary fibrosis, regorafenib, myofibroblasts, TGF-β1 signaling pathway

## Abstract

Idiopathic pulmonary fibrosis (IPF) is a fatal and age-related pulmonary disease. Nintedanib is a receptor tyrosine kinase inhibitor, and one of the only two listed drugs against IPF. Regorafenib is a novel, orally active, multi-kinase inhibitor that has similar targets to nintedanib and is applied to treat colorectal cancer and gastrointestinal stromal tumors in patients. In this study, we first identified that regorafenib could alleviate bleomycin-induced pulmonary fibrosis in mice. The in vivo experiments indicated that regorafenib suppresses collagen accumulation and myofibroblast activation. Further in vitro mechanism studies showed that regorafenib inhibits the activation and migration of myofibroblasts and extracellular matrix production, mainly through suppressing the transforming growth factor (TGF)-β1/Smad and non-Smad signaling pathways. In vitro studies have also indicated that regorafenib could augment autophagy in myofibroblasts by suppressing TGF-β1/mTOR (mechanistic target of rapamycin) signaling, and could promote apoptosis in myofibroblasts. In conclusion, regorafenib attenuates bleomycin-induced pulmonary fibrosis by suppressing the TGF-β1 signaling pathway.

## 1. Introduction

Idiopathic pulmonary fibrosis (IPF) belongs to the family of interstitial lung diseases and is a chronic, progressive, fibrotic interstitial lung disease of unknown cause that occurs primarily in the elderly [1,2]. The main pathological features of IPF are the massive accumulation of myofibroblasts and the excessive deposition of extracellular matrix in the lung interstitium, leading to the destruction of alveolar structure and interruption of gas exchange, after which patients have died of respiratory failure [3]. Its frequency is similar to that of gastric, brain, and testicular cancers, and the median survival time from diagnosis is 2.8 years [4]. The incidence of IPF has increased over time [5]. In Europe and North America, the estimated cases are 2.8 and 18.0 per 100,000 people per year, respectively [6,7]. Two drugs, nintedanib and pirfenidone, have been approved for the treatment of IPF, and can increase the mandatory vital capacity of patients to a certain extent; however, they have limited efficacy [8].

Transforming growth factor-β1 (TGF-β1) is the most critical profibrotic cytokine in the initiation and perpetuation of organ fibrosis [9], including IPF [10,11]. TGF-β1 plays a central role in regulating various processes of pulmonary fibrosis, such as epithelial injury, myo-fibroblast proliferation and differentiation, and extracellular matrix (ECM) production [12]. The conduction of TGF-β1 mainly depends on the typical Smad signaling pathway. Smad2 and Smad3 are phosphorylated by the TGF-β receptor, and then interact with Smad4 and translocate into the nucleus for transcriptional regulation [13,14]. Nonclassical or non-Smad signaling pathways are also important in TGF-β1 signaling, including p38 (P38 Mitogen-activated protein kinase), ERK (extracellular regulated protein kinases), JNK (c-Jun N-terminal kinase), PI3K (phosphatidylinositol 3 kinase), and Src (steroid receptor coactivator) family kinases [15]. The mechanistic target of rapamycin (mTOR) signaling pathway mediates a variety of critical cellular processes, including inflammation, cell cycle progression, autophagy, and metabolic and synthetic pathways [16]. Autophagy is a regulator of fibrogenesis of human primary atrial myofibroblasts induced by TGF-β1, and its inhibition prevents the conversion of cardiac fibroblasts to cardiac myofibroblasts [17,18]. When fibroblasts respond to TGF-β1 stimulation, canonical Smad signaling activates mTOR signaling [19].

Similar to nintedanib, regorafenib (RG) is an oral multi-kinase inhibitor that can effectively block multiple protein kinases of tumor angiogenesis, tumorigenesis, tumor metastasis, and tumor immunity, such as VEGFR (vascular endothelial growth factor receptor), TIE2 (endothelial tyrosine kinase Tie2), c-KIT (c-kit proto-oncogene), RET (RET proto-oncogene) and CSF1R (Colony-stimulating factor 1 receptor) [20,21]. This drug has been approved as a treatment of metastatic colorectal cancer (mCRC) and metastatic gastrointestinal stromal tumors (GISTs), and as a second-line treatment of hepatocellular carcinoma (HCC) [22]. Here, the therapeutic effect and antifibrotic mechanism of RG were first explored in a bleomycin (BLM)-induced pulmonary fibrosis model.

## 2. Results

### 2.1. RG Attenuates BLM-Induced Pulmonary Fibrosis in Mice

Whether RG alleviates BLM-induced pulmonary fibrosis in C57BL/6J mice was examined. We firstly explored the cytotoxicity and animal toxicity of RG, our experiments revealed that RG had not toxic effect on Mlg and NaCl-PPF cells and C57BL/6j mice (Appendix A). The mice orally received RG (15 mg/kg, 30 mg/kg) every day from day 7 to day 13. The results revealed that the fibrosis of mice was slowed after RG administration. RG markedly decreased the hydroxyproline content in the right lung and increased the body weight and percent survival (Figure 1A,B,E). Histological examination showed that RG repaired the structure of the alveolus and decreased the fibrotic percentage (Figure 1C,D). Masson staining was used to analyze the collagen level, and the results revealed a decline in collagen levels in lung sections (Figure 1C). In addition, the immunocytochemistry results revealed that RG suppressed the expression levels of α-SMA, Col 1, and Fn in lung tissues (Figure 1F). Pulmonary function is an important health indicator. The experimental results revealed that RG improved pulmonary function-related parameters, including forced vital capacity (FVC), forced expiratory volume in 0.1 s (FEV 0.1), forced expiratory volume in 0.3 s (FEV 0.3), expiratory resistance (Re), inspiratory resistance (Ri), and dynamic compliance (Cydn) (Figure 1G–L). Moreover, we repeated the effect of high dose 30 mg/kg RG at BLM-induced pulmonary fibrosis in mice, and the data showed similar result with the dose dependent experiment (Appendix A). In conclusion, RG sharply attenuated BLM-induced pulmonary fibrosis in mice, and could be applied to treat fibrosis in patients with IPF.

### 2.2. RG Suppresses TGF-β1-Induced Activation, ECM Accumulation, and Pulmonary Fibroblast Migration

Fibroblast overactivation is the main method of cells to accelerate fibrotic procession in patients with IPF [23]. RG attenuated the TGF-β1-induced expression of α-SMA and Col 1 in Mlg and NaCL-PPF cells (Figure 2A,B and Appendix AA,B), and RG also inhibited the α-SMA expression level in BLM-PPF cells instead of NaCl-PPF cells (Appendix A). Quantitative real-time PCR results revealed that RG decreased the TGF-β1-induced mRNA levels of α-SMA, Col 1a1, and FN in Mlg cells (Figure 2C). Immunofluorescence was also performed for α-SMA in Mlg cells, and revealed that RG can inhibit the TGF-β1-induced protein levels of α-SMA (Figure 2D). In addition, a cell migration experiment showed that RG inhibited lung fibroblast migration in Mlg and BLM–PPF cells (Figure 2E,F). In summary, RG inhibits TGF-β1-induced myofibroblast activation, ECM accumulation, and lung fibroblast migration.

### 2.3. RG Down-Regulates TGF-β1/Smad and TGF-β1/Non-Smad Signals in Pulmonary Fibroblasts

TGF-β1/Smad and TGF-β1/non-Smad signals play essential roles in pulmonary fibrotic progression in regulating fibroblast overaction. A TGF-β1/Smad3 luciferase reporter system was established to measure the inhibitory function of RG on this signal in CACA-NIH-3T3 cells. CAGA-NIH-3T3 cells treated with RG (0.0, 0.5, 1.0, 2.0, 4.0, 8.0, 16.0, and 32.0 μM) exhibited a dose-dependent decline of luciferase (Figure 3A). RG had a negative effect on the TGF-β1-induced phosphorylation of p-Smad3 (S423/425) and p-Smad2 (S476) in Mlg cells (Figure 3B). RG also inhibited the TGF-β1-induced phosphorylation of p-Akt (Ser473), p-Erk1 (Thr202), and p-Erk2 (Thr204) in Mlg cells, and BLM-PPF showed the same results (Figure 3C,D and Appendix AC,D). Therefore, RG suppressed TGF-β1/Smad and TGF-β1/non-Smad signaling, mainly by inducing the protein levels of p-Samd3, p-Smad2, p-Akt, and p-Erk1/2 in pulmonary fibroblasts.

### 2.4. RG Induces Autophagy in Pulmonary Fibroblasts Mainly by Inhibiting the TGF-β1/mTOR Signaling Pathway

Autophagy formation plays a key role in fibrotic treatment, and whether RG positively affects autophagic flux and the underlying mechanism were investigated. NaCl-PPF and BLM-PPF cells were isolated from the lung of mice. The results revealed that RG increased the expression level of LC3-II, Atg3, and Atg7, and decreased those of p62 in PPF cells (Figure 4A). Bafilomycin A1 (Baf A1) and chloroquine (CQ), two kinds of autophagy inhibitors, increased p62 protein expression levels in pulmonary fibroblasts, but RG reversed this effect (Figure 4B,C). GFP-LC3B and mCherry-GFP-LC3B plasmids were transfected into NIH-3T3 cells to detect autophagy flux, and RG increased the number of autophagosomes and autolysosomes (Figure 4D,E). mTOR signaling is a core regulator of autophagic flux. Hence, the efficiency of RG as an inhibitor of mTOR signaling was evaluated. RG attenuated the TGF-β1-induced phosphorylation of p-mTOR (S2248), p-ULK-1 (S757), p-p70S 6K (Thr389/412), and p-S6RP (S235/236) (Figure 4F). RG was administered to BLM-PPF cells and decreased the phosphorylation levels of p-mTOR (S2248), p-ULK-1 (S757), and p-S6RP(S235/236) (Figure 4G). In conclusion, RG promotes autophagy mainly by suppressing the TGF-β1/mTOR signaling pathway.

### 2.5. RG Promotes the Apoptosis of Myofibroblasts

Promoting the apoptosis of myofibroblasts can alleviate pulmonary fibrosis in patients with IPF. Flow cytometry is a prominent method for analyzing the effect of apoptosis. The results show that RG increased the number of instances of late apoptosis of activated fibroblasts induced by TGF-β1, but its effect on the early apoptosis of cells was not significant (Figure 5A). Increasing the expression of apoptosis-related proteins in myofibroblasts was beneficial for fibrotic progression. Our cellular results indicated that RG increased the TGF-β1-induced expression levels of cleaved PARP and cleaved-caspase 9 in Mlg cells, as well as cleaved PARP in BLM-PPF cells (Figure 5B,C). In summary, RG attenuated pulmonary fibrosis mainly by promoting the apoptosis of myofibroblasts.

### 2.6. RG Promotes Autophagy Formation and Inhibits TGF-β1/mTOR Signaling In Vivo

RG was used to verify the mechanism that alleviates pulmonary fibrosis in the in vitro cellular model and animal model. Hence, the following experiments mainly concentrated on the evaluation of RG’s in vivo effects. C57BL/6J mice were treated with RG (15 or 30 mg/kg) from days 7–14 after BLM was induced, and the lung tissues were isolated from mice as soon as possible, and extracted to analyze the effect of RG on autophagy formation and TGF-β1 signaling in vivo. Western blot analysis showed that RG suppressed the expression levels of p-Smad3 (Ser423/Ser412), p-p70S 6K (Thr389/412), and p62 in left lung tissues (Figure 6). These experimental results implied that RG promotes myofibroblast autophagy and inhibits myofibroblast activation and ECM production mainly via the TGF-β1 signaling.

## 3. Discussion

In this research, RG inhibited pulmonary fibrosis in vitro and in vivo. This drug can prevent myofibroblast activation and ECM production and promote autophagy. RG also promoted mouse lung function in the BLM-induced pulmonary fibrosis model. These findings indicate that RG improves lung fibrosis mainly by inhibiting myofibroblast activation and ECM production and promoting autophagy, suggesting that this drug can prevent fibrotic progression.

TGF-β1 is a key cytokine extensively involved in the development of organ fibrosis; it regulates the activation, migration, and ECM production of myofibroblasts through the TGF-β/Smad and non-Smad signaling pathways [9,10,11,24]. The current experiment results suggested that RG inhibited TGF-β1-induced fibroblast activation and ECM production mainly by suppressing the TGF-β1/Smad and non-Smad signaling pathways in vitro and in vivo. RG also inhibited the migration of fibroblasts induced by TGF-β1.

Autophagy is closely related to pulmonary fibrosis. In the lung tissues of patients with IPF, the expression level of autophagic protein p62 is higher than the normal value [25], and mTOR expression is closely related to the fibrosis score and decreased lung function, suggesting a possible relationship to the prognosis of pulmonary fibrosis [26]. In a BLM mouse model of pulmonary fibrosis, BLM binding with ANXA2 impedes transcription factor EB (TFEB)-induced autophagic flux [27]; the inhibition of profibrotic cytokines induces autophagy flux and decreases the production of collagen to alleviate pulmonary fibrosis [28]. RG directly stabilizes phosphoserine aminotransferase 1 to trigger autophagy initiation and inhibit RAB11A-mediated autophagosome–lysosome fusion, leading to fatal autophagy arrest in GBM cells [29]. In this study, RG decreased the expression of p62 and increase the formation of autophagy flux. p62 is a vital regulator of autophagy, which is mainly the result of p62 and LC3 interactions [30]. Whether RG has destructive effects on the affinity between p62 and LC3 is unclear. RG can also inhibit the TGF-β1-induced activation of mTOR signaling in lung fibroblasts, and exhibited the same results in a BLM-induced lung fibrosis model.

RG also promotes the TGF-β1-induced apoptosis of myofibroblasts. Fibroblasts in IPF patients have some characteristics of overactivation and anti-apoptosis, and limiting both of them can significantly remit the progression of pulmonary fibrosis [31]. A previous study showed that TGF-β1 suppressed interleukin-1β (IL-1β)-induced apoptosis of myofibroblasts in rat lungs, mainly by increasing the expression of the regulatory proteins Bax and Bcl-2 [32]. Current results indicate that RG can increase the activation of apoptosis-related proteins and the ratio of apoptotic cells in fibroblasts induced by TGF-β1.

In conclusion, RG attenuated BLM-induced pulmonary fibrosis in mice, inhibited myofibroblast activation and migration, and induced myofibroblast autophagy and apoptosis to downregulate ECM accumulation by suppressing the TGF-β1/Smad, TGF-β1/non-Smad, and mTOR signaling pathways. RG is a small, high-activity molecule in cancer treatment that might have a beneficial effect on improving fibrotic progression in patients with IPF. However, the specific targets of RG for TGF-β1/Smad signaling, TGF-β1/non-Smad signaling, and mTOR signaling, as well as the coordination between these signaling pathways, remain unclear and require further research.

## 4. Materials and Methods

### 4.1. Ethics Statement

All animal care and experimental procedures complied with the guidelines approved by the Institutional Animal Care and Use Committee (IACUC) of Nankai University (project code: SYXK 2019-0001, date of approval: 1 November 2019).

### 4.2. Animals

For this experiment, 6- to 8-week-old male C57BL/6J mice were purchased from Charles River (Beijing, China), and housed and cared for in a pathogen-free facility at Nankai University Experimental Animal Center under controlled temperature (22 °C–26 °C), a 12 h light–dark cycle, and relative humidity (60% ± 2%), with free access to food and water. Animal studies were reported in compliance with the ARRIVE guidelines. BLM (Medicine Co., Tokyo, Japan) was dissolved in saline (Sangon, Shanghai, China) and given through intratracheal instillation (2 U). Control mice received intratracheal instillation of saline only. The 50 mice were randomly categorized into the following five groups (*n* = 10 per group): control group, BLM (2 U) group, BLM + nintedanib (100 mg/kg), BLM + RG (15 mg/kg), and BLM + RG (30 mg/kg). The mice were orally exposed to nintedanib (Macklin, Shanghai, China) and Regorafenib (Meilun Biotechnology, Dalian, China) once a day for days 7–13, and were sacrificed on day 14. Lung tissues were harvested for the following experiments to evaluate the degree of pulmonary fibrosis.

### 4.3. Isolation Protocol of Primary Pulmonary Fibroblasts (PPFs)

PPF cells were isolated from NaCl and BLM-treated C57BL/6J mice, as mentioned earlier [33]. Briefly, the lungs were lavaged three times with 1 mL PBS and digested with 2.5 mg/mL DispaseⅡ (Roche Diagnostics, Indianapolis, IN, USA) and 2.5 mg/mL Collagenase Type 4 (Worthington Biochemical, Freehold, NJ, USA) for 30 min in 37 °C, and then centrifuged. The cell pellet was resuspended in Dulbecco’s modified eagle medium (DMEM) containing 10% fetal bovine serum (FBS) and cultured in 5% CO_2_ at 37 °C in a humidified atmosphere. Cells at passages 2–5 were used for various assays.

### 4.4. Cell Culture

The mouse lung fibroblast (Mlg), mouse embryonic fibroblast (NIH-3T3), and CAGA-NIH3T3 cell lines were kindly provided by Professor Wen Ning, Nankai University, and were maintained in DMEM (Solarbio, Beijing, China) containing 10% FBS (Gibico, Carlsbad, CA, USA) and 1% penicillin–streptomycin (Gibico, Carlsbad, CA, USA). Mouse primary pulmonary fibroblast (PPF) cells were obtained from NaCl or BLM-induced mice on day 14. Mlg, NIH-3T3, and PPF cells in DMEM with 1‰ FBS were stimulated with TGF-β1 (human TGF-beta1-Mammalian; Lianke Biotechnology, Hangzhou, China) or RG (dissolved into DMSO). All cells were cultured in a constant temperature incubator with 5% CO_2_ at 37 °C.

### 4.5. Western Blot

RIPA buffer (Beyotime, Shanghai, China) with protease inhibitor cocktail (Cell Signaling Technology, Beverly, MA, USA) was used to prepare the lysates, and protein concentrations were measured using a BCA protein assay (Beyotime, Shanghai, China). Next, SDS-PAGE was performed to separate proteins, followed by transferring proteins into polyvinylidene difluoride (PVDF) membranes (Roche Diagnostics, Indianapolis, IN, USA) and blocked with 5% skimmed milk in TBS-T. After incubation with primary and secondary antibodies, immunoreactivity was detected by ECL (Affinity Biosciences, OH, USA). HRP-labelled, goat anti-rabbit IgG (H+L) (Cell Signaling Technology, Beverly, MA, USA) was used as the secondary antibody to detect the primary antibodies from rabbit, and HRP-labelled, goat anti-mouse IgG (H+L) (Cell Signaling Technology, Beverly, MA, USA) antibodies were used as the secondary antibodies to detect the primary antibodies from mice. The relative density was analyzed by ImageJ. The following primary antibodies (Table 1) were used to detect the protein expression levels.

### 4.6. Real-Time Quantitative PCR

Total RNA was extracted from adherent cells or tissue using TRIZOL (Thermo Scientific Inc, Waltham, MA, USA) according to the manufacturer’s instructions. Gene primers (Table 2) of α-SMA (α-smooth muscle actin), Fn (Fibronectin) and Col1 A1 (Collagen I A1) were purchased from Qingke Biological Technology (Beijing, China). Transcription was performed using the Reverse SYBR Select Master Mix kit, according to the instructions (Tiangen, Beijing, China), followed by fluorescence quantitative real-time PCR (Yeasen, Shanghai, China). PCR amplification was carried out for 40 cycles at a melting temperature of 95 °C for 15 s and an annealing temperature of 60 °C for 1 min.

### 4.7. Luciferase Assay

CAGA-NIH-3T3 cells were established by stably transfecting 12 copies of the Smad3 binding sequence (CAGA) in front of the luciferase reporter gene upstream of the TGF-β promoter. The fold change in luciferase represents the inhibitory effect of RG on TGF-β signaling. The cells were seeded in 96-well plates at 5000 cells per well and cultured overnight to attach. After serum starvation for 24 h, TGF-β1 (5 ng/mL) and/or RG (0–32 μM) was added and incubated for 18 h. The cells were then harvested, and the lysates were used to determine the luciferase activity with a luciferase assay system (Promega, Madison, Wisconsin, USA), as described by the manufacturer. Total light emission was measured with a Luminoskan Ascent Reader System (Thermo Scientific Inc, Waltham, MA, USA).

### 4.8. Wound-Healing Assays

The Mlg cell line and PPF cells from C57/BL/6J mice were inoculated in six-well plates for the wound healing assay, and were scraped to form a wound by using 200 µL sterile pipette tips. These cells were cultured in the presence of RG (2 or 4 µM) and/or TGF-β1 (5 ng/mL) in DMEM with 1‰ FBS. Images of the cells were obtained at a series of time points (0, 12, 24, and 36 h) by using a light microscope (Nikon, Tokyo, Japan).

### 4.9. Hematoxylin–Eosin (H&E) Staining

Mouse left lungs were fixed overnight in 10% formalin and embedded in paraffin. Lung sections (5 µm) were prepared for hematoxylin–eosin staining and Masson’s trichrome staining (Solarbio, Shanghai, China). Images were collected using a fluorescence microscope (Nikon, Japan) and opened in Image-Pro Plus Version 6.0 (Media Cybernetics Inc, Bethesda, MD, USA). Quantification of pulmonary fibrosis was performed as described previously [34].

### 4.10. Immunohistochemistry

Immunohistochemical (IHC) analyses were performed to analyze α-smooth muscle actin (α-SMA), collagen 1 (Col 1), and fibronectin (FN) expression levels in mouse lung sections by using the UltraSensitiveTM SP (Mouse/Rabbit) IHC Kit and DAB Kit (Maxim, Fuzhou, China). In brief, the tissue sections were pretreated in a microwave, blocked, and incubated with primary antibodies for one night at 4 °C, and stained with DAB and hematoxylin. The results were captured using a fluorescence microscope (Nikon, Tokyo, Japan).

### 4.11. Immunofluorescence

For immunofluorescence, after performing the dosing process, cells were fixed in 4% paraformaldehyde for 20 min, and then 0.2% Triton X-100 was used for permeabilization, using 5% BSA to block nonspecific protein sites. The cells were then incubated with α-SMA antibody at 4 °C overnight. The next day, the primary antibody was removed, and the cells were washed with 1× PBS-T three times. Fluorescein (FITC) AffiniPure goat anti-mouse IgG (H+L) (Jackson Immunoresearch, West Grove, PA, USA) was added and incubated for 1.5 h in the dark. Nuclei were stained with DAPI (Solarbio, Beijing, China). Cells were imaged with a TCS SP8 confocal microscope (Leica). For microtubule-associated protein 1 light chain 3 (LC3)-B plasmid transfection, according to the supplier’s instructions (Sino Biological, Beijing, China), 1.5 mg GFP-LC3 and 2 mg Cherry-GFP-LC3 plasmids were transfected into NIH-3T3 cells using PEI (Thermo Scientific Inc., Waltham, MA, USA). After RG and TGF-β1 (5 ng/mL) treatment for 24 h, the cells were fixed, the cellular nuclei were stained, and pictures were taken based on the above similar steps.

### 4.12. Flow Cytometric Analysis of Apoptosis

Mlg cells were seeded in six-well plates and left for 24 h in an incubator to resume exponential growth. The extent of apoptosis was measured using an annexin V–FITC apoptosis detection kit (Beyotime, Shanghai, China), following the manufacturer’s instruction. The cells were exposed to RG (2 or 4 µM) and/or TGF-β1 (5 ng/mL) for 24 h, collected, washed with PBS twice, gently resuspended in annexin V binding buffer, incubated with annexin V-FITC/PI in the dark for 15 min, and finally analyzed by flow cytometry (BD Biosciences, San Diego, CA, USA).

### 4.13. Hydroxyproline Assay

A conventional hydroxyproline method was used to measure the collagen content in the mouse right lungs. The right lungs were dried, acid hydrolyzed, and adjusted to pH 6.5–8.0. Chloramine-T (MERYER, Shanghai, China) spectrophotometric absorbance, as previously described, was used for hydroxyproline analysis [35]. Finally, the hydroxyproline level in the mouse right lung was measured by a spectrophotometer at 550 nm.

### 4.14. Evaluation of Pulmonary Function

The mice were anesthetized with 1% pentobarbital sodium in NaCl (i.p.) and transferred to a plethysmography chamber for pulmonary function analyses using the Anires2005 system (Biolab, Beijing, China). This system automatically calculates and displays pulmonary function parameters, including forced vital capacity (FVC), forced expiratory volume in 0.1 s (FEV0.1), forced expiratory volume in 0.3 s (FEV0.3), dynamic compliance (Cydn), inspiratory resistance (Ri), and expiratory resistance (Re).

### 4.15. Statistical Analysis

GraphPad Prism 6.0 was used for Statistical analysis. One-way ANOVA, Pearson correlation, and linear regression methods were used to analyze significant differences and concordance. Values are the means ± standard error of the mean (SEM), and significance was described as follows: # represent the difference between control/NaCl and TGF-β1/BLM-treated group, and * represents the difference between the TGF-β1/BLM-treated and the RG-treated group: *, *p* <0.05; **, *p* < 0.01; ***, *p* < 0.001. ^#^, *p* <0.05; ^##^, *p* < 0.01; ^###^, *p* < 0.001, NS: not significant.

## Figures and Tables

**Figure 1 ijms-22-01985-f001:**
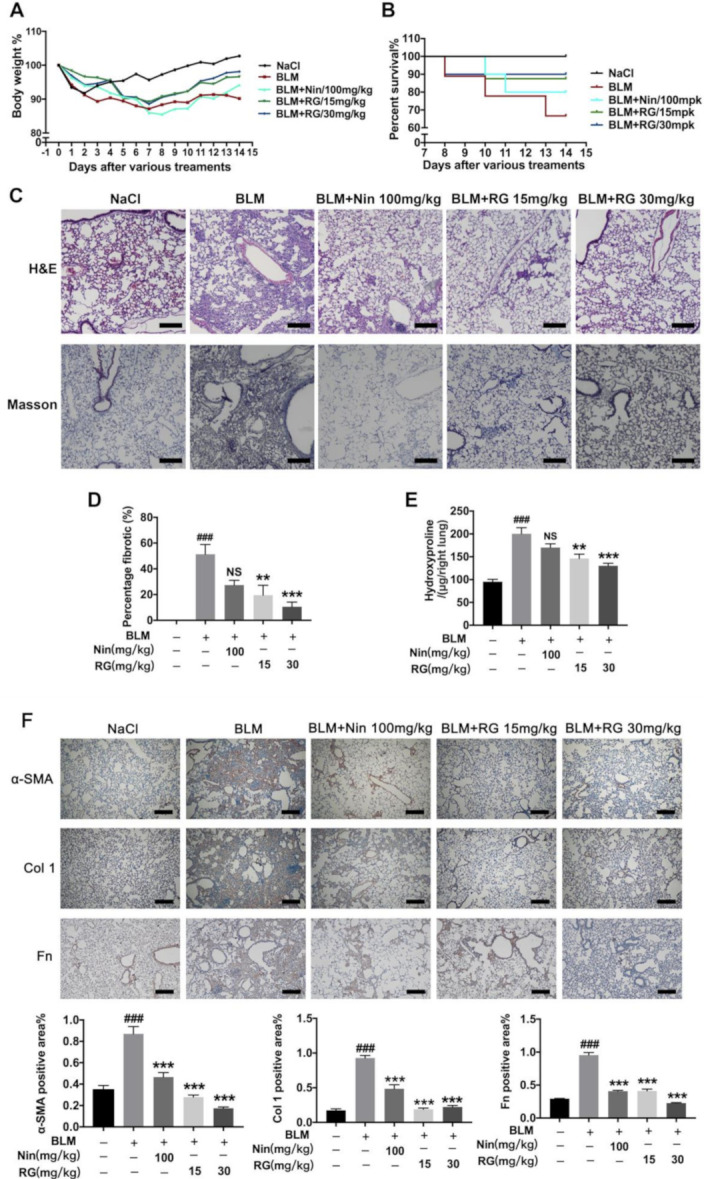
Regorafenib attenuated bleomycin (BLM)-induced pulmonary fibrosis in mice. (**A**) Regorafenib (15 mg/kg, 30 mg/kg) and nintedanib (100 mg/kg) were given orally once a day from days 7–13 after BLM treatment, lungs were harvested at day 14, and body weight loss was measured every day; (**B**) Percentages of surviving mice were plotted from days 7–14 after bleomycin treatment; (**C**) Representative images of hematoxylin–eosin (H&E) and Masson staining of lung tissue sections. Scale bars: 50 µM; (**D**) Percentages of fibrotic area in lung tissues; (**E**) Hydroxyproline contents in right lung tissues; (**F**) Immunohistochemistry of lung sections was used to analyze the expression levels of α-smooth muscle actin (α-SMA), collagen 1 (Col 1), and fibronectin (Fn). Quantitative analysis is shown below. Scale bars: 50 μm; (**G**–**L**) Pulmonary function parameters, such as forced vital capacity (FVC), forced expiratory volume in 0.1 s (FEV0.1), forced expiratory volume in 0.3 s (FEV0.3), expiratory resistance (Re), inspiratory resistance (Ri), and dynamic compliance (Cydn). Data in (**A**,**B**) and (**D**–**L**) are means ± standard error of mean (SEM), *n* = 6; # *p* < 0.05, ## *p* < 0.01, ### *p* < 0.001, * *p* < 0.05, ** *p* < 0.01, *** *p* < 0.001 (one-way ANOVA), NS: not significant.

**Figure 2 ijms-22-01985-f002:**
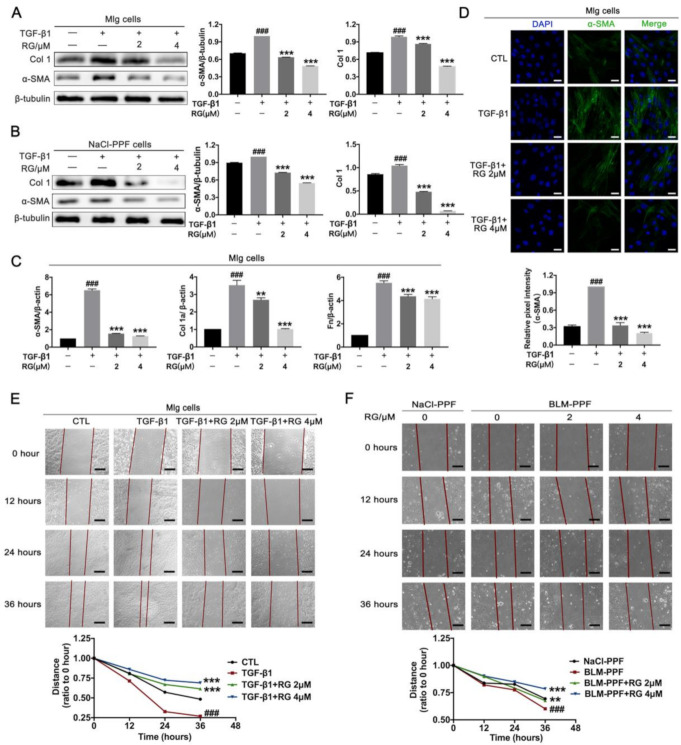
Regorafenib suppresses transforming growth factor (TGF)-β1-induced activation, extracellular matrix (ECM) accumulation, and the migration of pulmonary fibroblasts. (**A**,**B**) Mouse lung fibroblast (Mlg) or NaCl-primary pulmonary fibroblast (PPF) cells were exposed to TGF-β1(5 ng/mL) and/or regorafenib (RG) (2 µM, 4 µM) for 24 h to detect the expression levels of α-SMA and Col 1 by Western blotting. Densitometric analyses are shown; (**C**) Mlg cells were exposed to TGF-β1 (5 ng/mL) and/or RG (2 µM, 4 µM) for 24 h, and quantitative real-time PCR was used to detect the mRNA levels of α-SMA, Col 1a, and Fn; (**D**) RG (2 µM, 4 µM) was exposed to Mlg cells for 24 h, and the α-SMA expression level was detected by immunofluorescence. Relative pixels intensity is shown. Scale bars: 50 μm; (**E**) Mlg cells were treated with TGF- β1 and/or RG (2 μM, 4 μM) for 0, 12, 24, or 36 h. Distance analyses are shown below. Scale bars: 100 μm; (**F**) The PPF cells isolated from NaCl-treated and BLM-treated mice were incubated RG (2 μM, 4 μM) for a series of time points (0, 12, 24, 36 h). Distance analyses are shown below. Scale bars: 100 μm; Data in (**A**–**F**) are means ± standard error of mean (SEM), and β-tubulin was used as a loading control. ### *p* < 0.001, ** *p* < 0.01, *** *p* < 0.001 (one-way ANOVA), NS: not significant.

**Figure 3 ijms-22-01985-f003:**
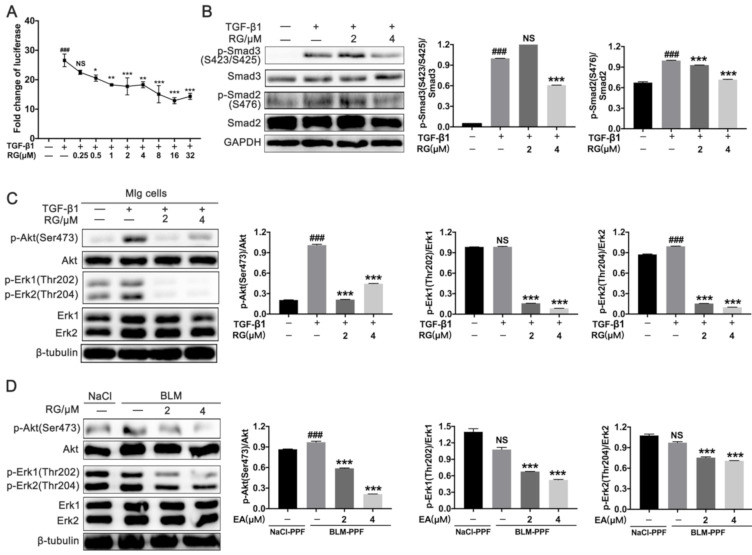
Regorafenib down-regulates TGF-β1/Smad and TGF-β1/non-Smad signals in pulmonary fibroblasts. (**A**) CAGA- mouse embryonic fibroblast (NIH-3T3) cells were exposed to TGF-β1 (5 ng/mL) or a series concentration (0–32 µM) in serum-free medium for 18 h; (**B**) Mlg cells were treated with TGF-β1 (5 ng/mL) and/or RG (2 µM, 4 µM) for 30 min, and Western blot was used to detect Smad3, Smad2, and their phosphorylation expression levels. Densitometric analysis are shown beside; (**C**) Mlg cells were incubated with RG (2 μM, 4 μM) and/or TGF-β1 (5 ng/mL) for 12 h to analyze the Erk1/2 and Akt and their phosphorylation levels by Western blotting. Densitometric analysis are shown beside; (**D**) BLM-PPF cells were incubated with RG (2 μM, 4 μM) for 12 h to analyze Erk1/2 and Akt and their phosphorylation levels by Western blotting. Densitometric analysis are shown beside. β-tubulin or GAPDH were used as a loading control. Data in (**A**–**D**) are means ± standard error of mean (SEM); ### *p* < 0.001, * *p* < 0.05, ** *p* < 0.01, *** *p* < 0.001 (one-way ANOVA), NS: not significant.

**Figure 4 ijms-22-01985-f004:**
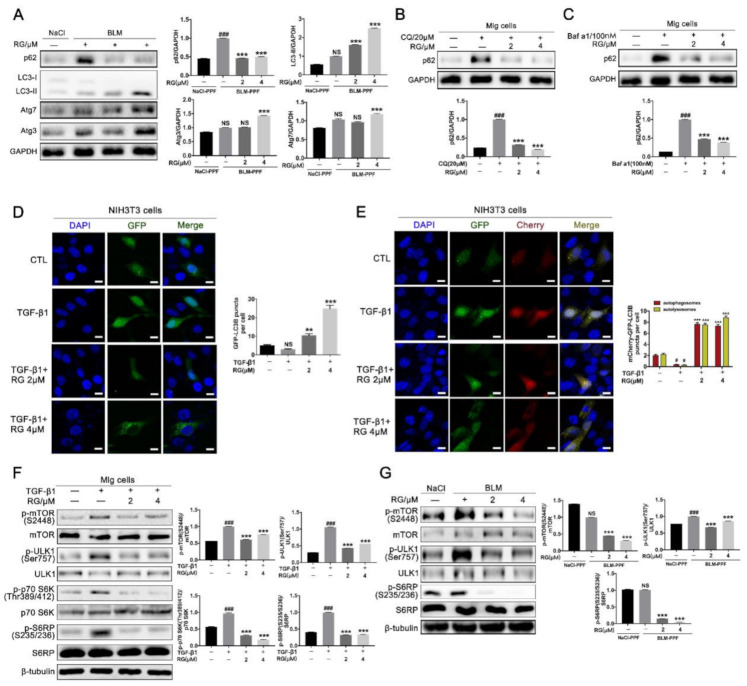
Regorafenib induces the formation of autophagy by inhibiting the TGF-β1/mTOR signal pathway. (**A**) BLM-PPF cells were treated with RG (2 μM, 4 μM) for 24 h, and the expression levels of autophagy-related proteins (Atg3, Atg7, p62, and LC3-II/I) were detected. Densitometric analysis are shown beside; (**B**) Mlg cells were exposed to chloroquine (CQ) and/or RG (2 μM, 4 μM) for 24 h, and Western blot technology was used to detect the p62 protein expression level; (**C**) Mlg cells were exposed to bafilomycin A1 (Baf A1) and/or RG (2 μM, 4 μM) for 24 h. Densitometric analysis are shown below; (**D**,**E**) The plasmids of GFP-LC3B (green fluorescent protein-microtubule-associated protein 1 light chain 3B) and mCherry-GFP-LC3B (mCherry fluorescent protein-green fluorescent protein-microtubule-associated protein 1 light chain 3B) was transferred into NIH-3T3 cells, and these cells subsequently were incubated with RG (2 μM, 4 μM) and/or TGF-β1 (5 ng/mL) for 12 h. DNA was counterstained with DAPI (blue). Scale bars: 50 μm; (**F**) Mlg cells were incubated with RG (2 μM, 4 μM) and/or TGF-β1 (5 ng/mL) for 12 h, and the expression levels of mTOR (mechanistic target of rapamycin), ULK-1 (protein kinase ULK1/autophagy-related protein 1), p70 S6K (p70 S6 kinase), and S6RP (S6 ribosomal protein), and their phosphorylations were detected. Densitometric analysis are shown beside; (**G**) BLM-PPF cells were explored to RG (2 μM, 4 μM) for 12 h, and Western Blot were used to detect the expression levels of mTOR, ULK-1, and S6RP, and their phosphorylations. Densitometric analysis are shown beside. Data in (**A**–**G**) are means ± standard error of mean (SEM), and GAPDH or β-tubulin were used as a loading control. # *p* < 0.05, ### *p* < 0.001, ** *p* < 0.01, *** *p* < 0.001(one-way ANOVA). NS: not significant.

**Figure 5 ijms-22-01985-f005:**
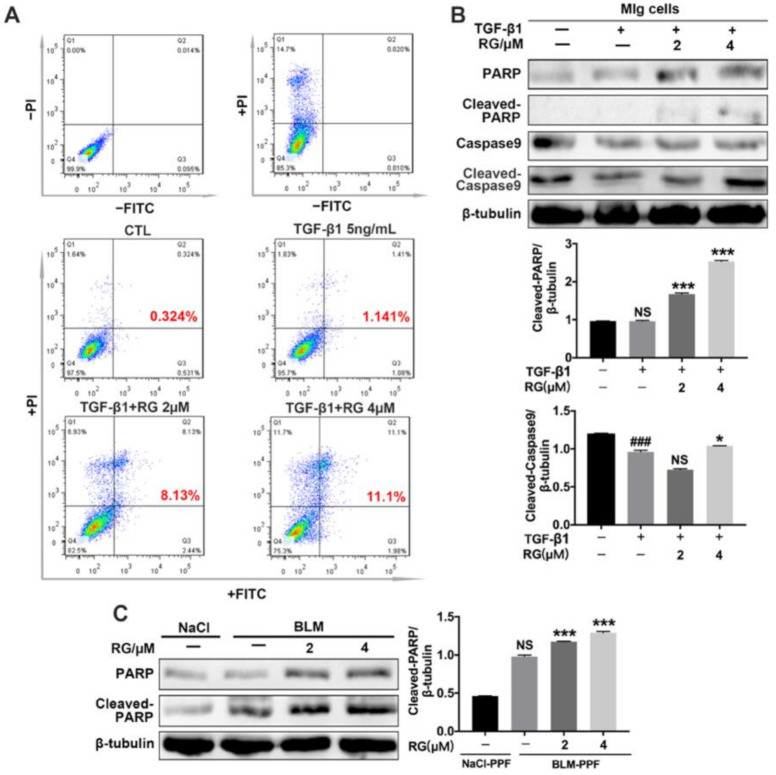
Regorafenib promotes TGF-β1-induced apoptosis of myofibroblasts. (**A**) Mlg cells were treated with TGF-β1 and/or RG (2 μM, 4 μM) for 24 h. Annexin V/PI staining was subsequently performed to assess early-apoptosis cells, late-apoptosis cells, and necrosis cells; 5 × 10^3^ cells were analyzed by flow cytometry; (**B**) Mlg cells were treated with TGF-β1 and/or RG (2 μM, 4 μM) for 24 h, and expression levels of Caspsed9 and PARP and their activation were detected by Western blotting. Densitometric analysis are shown below; (**C**) RG (2 μM, 4 μM) was exposed to BLM-PPF cells for 24 h, and the expression levels of PARP and its activation were detected by Western blot. Densitometric analyses are shown. Data in (**B**,**C**) are means ± standard error of mean (SEM), ### *p* < 0.001, * *p* < 0.05, *** *p* < 0.001 (one-way ANOVA), NS: not significant.

**Figure 6 ijms-22-01985-f006:**
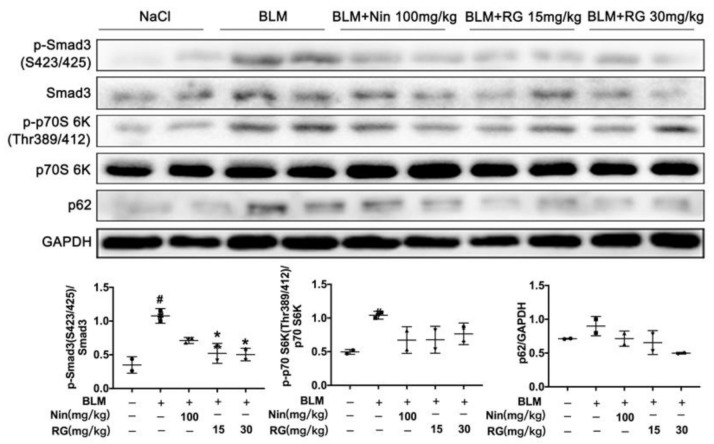
RG promotes autophagy formation and inhibits TGF-β1/mTOR signaling in vivo. The mice were orally treated with RG (15 mg/kg, 30 mg/kg) from day 7 to day 13 after BLM administration, and lung homogenization was used to analyze the expression levels of p-Smad3 (Ser423/Ser425), p-p70 S6K (Thr389/412), and p62 by Western blotting. Densitometric analysis is shown. # *p* < 0.05, * *p* < 0.05 (one-way ANOVA), NS: not significant.

**Table 1 ijms-22-01985-t001:** The list of primary antibodies.

Antibody	Company and Item No.	Antibody	Company and Item No.
GAPDH	Affinity, AF7021	Beclin1	CST, 3495S
β-tubulin	Affinity, T0023	LC3A/B	CST, 12741S
α-SMA	Affinity, BF9212	mTOR	Abcam, ab32028
Collagen I	Affinity, AF7001	p-mTOR (S2448)	Abcam, ab109268
p-Smad3 (S423/425)	Affinity, AF8315	S6 ribosomal protein (S6RP)	Affinity, AF7831
Smad3	Affinity, AF6362	p-S6 ribosomal protein (S235/S236)	CST, 4858T
p-Smad2(S467)	Affinity, AF3449	p-ULK1( Ser757)	CST, 14202T
Smad2	Affinity, AF6449	ULK1	CST, 8054T
Akt	SANTA CRUZ, sc-56878	p-p70 S6K (Thr289/412)	Affinity, AF3228
p-Akt (Ser473)	SANTA CRUZ, sc-514032	p70 S6K	Affinity, AF6226
Erk1/2	SANTA CRUZ, sc-514302	PARP	CST, 9532T
p-Erk1/2 (Thr202/Thy204)	SANTA CRUZ, sc-81492	Cleaved-PARP	CST, 5625T
Atg7	CST, 8558T	Caspase9	CST, 9009T
P62	Proteintech, 8420-1-AP	Cleaved-Caspase9	CST, 9504T
Atg3	CST, 8089T		

**Table 2 ijms-22-01985-t002:** The list of gene primers.

Gene (Mouse)	Forward Primer Sequence (5′-3′)	Reverse Primer Sequence (5′-3′)
α-SMA	GCTGGTGATGATGCTCCCA	GCCCATTCCAACCATTACTCC
Fn	GTGTAGCACAACTTCCAATTACGAA	GGAATTTCCGCCTCGAGTCT
Col1 A1	CCAAGAAGACATCCCTGAAGTCA	TGCACGTCATCGCACACA
β-actin	AGGCCAACCGTGAAAAGATG	AGAGCATAGCCCTCGTAGATGG

## Data Availability

Not applicable.

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
