# Peer review of "Regorafenib-Attenuated, Bleomycin-Induced Pulmonary Fibrosis by Inhibiting the TGF-β1 Signaling Pathway"

_ijms, 2021, doi:10.3390/ijms22041985_

Round 1

Reviewer 1 Report

In the current manuscript, the authors describe their findings regarding the activity of the TKI regorafenib in bleomcyin-induced fibrosis in vivo and in cell culture with respect to, inter aliter, TGFb pathway activity, autophagy and apoptosis. On the whole, the paper is well written, and the dat are well presented, but the text will require additional redacting to improve syntax and grammar.

The conclusions are heavily reliant on the interpretation of the results with respect to the inhibitory activity of regorafenib on TGFb signalling. In many of the in vitro experiments, there does indeed seem to be a remarkable effect of addition of the compound, but for all of these, a control with compound but lacking TGF, is absent, thus making interpretation of the data with respect to a TGFb-specific effect difficult, especially when levels of the read-out parameter are (sometimes significantly) below negative control (I.e. TGF -, RG -) levels. Off-target / non-TGFb-related or perhaps even completely non-specific effects can therefore not be excluded, and at least for those experiments, a relevant control (TGFb -, RG +) should be added (esp. Fig 2b, Fig. 3C+D).  

Author Response

Reviewer #1: In the current manuscript, the authors describe their findings regarding the activity of the TKI RG in bleomcyin-induced fibrosis in vivo and in cell culture with respect to, inter alter, TGF-β1 pathway activity, autophagy and apoptosis. General concerns:

Comment 1: The text will require additional redacting to improve syntax and grammar.

Reply 1: We thank the reviewer’s suggestion. We have modified the syntax and grammar of our manuscript.

Comment 2: In many of the in vitro experiments, there does indeed seem to be a remarkable effect of addition of the compound, but for all of these, a control with compound but lacking TGF-β1, is absent, thus making interpretation of the data with respect to a TGF-β1-specific effect difficult, especially when levels of the read-out parameter are (sometimes significantly) below negative control (i.e. TGF-β1, RG-) levels. Off-target/non-TGF-β1-related or perhaps even completely non-specific effects can therefore not be excluded, and at least for those experiments, a relevant control (TGF-β1, RG +) should be added (esp. Fig 2b, Fig. 3C+D).

Reply 2: We thank the reviewer’s suggestion. We have supplemented the effect of RG on Mlg and NaCl-PPF cells without TGF-β1 treatment. Our results showed that RG (2, 4µM) have no significant impacts on related protein expression of normal fibroblasts. For example, Mlg and NaCl-PPF cells were treated with RG for 24 hours, and the expression levels of α-SMA and Col 1 didn’t change (Fig.S1 A-B). In addition, RG was incubated with fibroblasts for 12 hours, the p-Erk1/2(Thr202/204) and p-Akt (Ser473) expression levels haven’t change either (Fig.S1 C-D).

Fig.S1 RG suppresses TGF-β1-induced the activation and TGF-β1-non-Smad pathway.

(A-B) Mlg or NaCl-PPF cells were exposed to TGFβ1(5ng/mL) and/or RG (2µM, 4µM) for 24 h to detect the expression levels of α-SMA, Col 1 by Western blot. Densitometric analysis were showed beside. (C) Mlg cells were incubated with RG (2μM, 4μM) and/or TGF-β1(5ng/mL) for 12 hours to analyze the Erk1/2, Akt and its phosphorylation expression levels by Western blot. Densitometric analysis were showed beside. (D) BLM-PPF cells were incubated with RG (2μM, 4μM) for 12 hours to analyze the Erk1/2, Akt and its phosphorylation expression levels by Western blot. Densitometric analysis were showed beside. β-tubulin or GAPDH were used as a loading control. Data in (A-D) are means ± Standard Error of Mean (SEM), *P<0.05, **P<0.01, ***P<0.001 (one-way ANOVA), NS: nonsignificant.

Reviewer 2 Report

In this MS, Li et al have assessed the therapeutic potential of Regorafenib (RG) in pulmonary fibrosis using invivo and invitro models. An extensive molecular characterization experiments to demonstrate the efficacy at cellular and molecular level is the strength of this MS, while the lack of appropriate controls dampens the enthusiasm. Here are the comments –

  1. Authors have set the premise that RG could be a possible anti fibrotic therapy for the treatment of IPF. However, the substantiating data are missing. For instance, the stand alone experiments to show the effects of RG on WT animals in the invivo or invitro is very important. Authors need to add up those experiments.
  2. BLM experiment is done on a very young animals, while the IPF is an ageing disorder. How well, the authors will be able to correlate this finding in the clinical set up?
  3. Authors may explain the BLM dose in terms of units, which is the standard way of explanation.
  4. Authors should explain the isolation protocol of PPF. Also, are the cells validated for the expression of markers? Please show all details in the MS (probably in supplements). This data is very important!
  5. TGFb1 experiments – What are the cell culture conditions maintained for this experiment? Like serum levels and how long TGF treatment? These are key information, and they are missing.
  6. Statistics – Use of t-test is totally inappropriate. For treatments with more than 2 groups, authors should be doing one way ANOVA, followed by post-hoc tests. Further, SEM is fine for invitro experiments, instead of SD. Authors are required to re-analyze all data for ANOVA and SEM
  7. As explained in (1), stand-alone effect of RG is required for invivo and invitro. Since authors have started the RG treatment on Day 8, a small animal experiment is do-able (for 7 days of treatment). This is a must control experiment. In ideal condition, authors should have set up a dose dependent experiment first, choose a dose and then repeat the experiment with the chosen dose with and out RG for with and without BLM and go for a 2-way ANOVA.
  8. What is the effect of RG in fibroblasts (MIg and PPF) in the context of proliferation or cytotoxicity? This data is very critical. Secondly, what is the correlation between invivo and invitro dose?

Author Response

Reviewer #2: In this MS, Li et al have assessed the therapeutic potential of RG (RG) in pulmonary fibrosis using in vivo and in vitro models. An extensive molecular characterization experiments to demonstrate the efficacy at cellular and molecular level is the strength of this MS, while the lack of appropriate controls dampens the enthusiasm. Here are the comments:

Comment 1: Authors have set the premise that RG could be a possible anti fibrotic therapy for the treatment of IPF. However, the substantiating data are missing. For instance, the stand-alone experiments to show the effects of RG on WT animals in the in vivo or in vitro is very important. Authors need to add up those experiments.

Reply 1: We thank the reviewer’s comment. We have tested the effect of RG on fibroblasts in vitro and WT animals in vivo as soon as we got the reviewer’s comments. We have successfully completed cytotoxicity and animal toxicity tests. The in vitro results indicated that 2µM and 4µM RG didn’t have cytotoxicity on Mlg and NaCl-PPF cells (Fig.S2 A-B). Moreover, RG didn’t influence the expression of biomarker proteins in normal fibroblasts, such as α-SMA, Col1, p-Akt and p-Erk1/2 (Fig.S1 A-D). The in vivo tests were used 15mg/kg, 30mg/kg, 150mg/kg and 300mg/kg RG to orally given C57BL/6 mice for 7 days, and the results showed that RG has no effects on the weight and the appearance of main organs (heart, liver, spleen, lung, kidney) in each group (Fig.S2 C-E).

Fig.S1 RG suppresses TGF-β1-induced the activation and TGF-β1-non-Smad pathway.

(A-B) Mlg or NaCl-PPF cells were exposed to TGFβ1(5ng/mL) and/or RG (2µM, 4µM) for 24 h to detect the expression levels of α-SMA, Col 1 by Western blot. Densitometric analysis were showed beside. (C) Mlg cells were incubated with RG (2μM, 4μM) and/or TGF-β1(5ng/mL) for 12 hours to analyze the Erk1/2, Akt and its phosphorylation expression levels by Western blot. Densitometric analysis were showed beside. (D) BLM-PPF cells were incubated with RG (2μM, 4μM) for 12 hours to analyze the Erk1/2, Akt and its phosphorylation expression levels by Western blot. Densitometric analysis were showed beside. β-tubulin or GAPDH were used as a loading control. Data in (A-D) are means ± Standard Error of Mean (SEM), *P<0.05, **P<0.01, ***P<0.001 (one-way ANOVA), NS: nonsignificant.

Fig.S2 RG had not toxic effect on Mlg and NaCl-PPF cells and C57BL/6 mice.

(A-B) Mlg and NaCl-PPF cells were incubated with RG (0, 0.125, 0.5, 1, 2, 4, 8, 16, 32, 64, 128, 256 and 512µM) for 24 h, and MTT assays were used to analyze the cytotoxicity effect of RG on the cells. (C) RG (15mg/kg, 30mg/kg, 150mg/kg and 300mg/kg) were given orally once a day for 7 days in normal mice, and the weight changes were shown from day 1 to day 7. (D) The appearance of mouse viscera. (E) The appearance of five main organs (heart, liver, spleen, lung, kidney).

Comment 2: BLM experiment is done on a very young animals, while the IPF is an ageing disorder. How well, the authors will be able to correlate this finding in the clinical set up?

Reply 2: We thank the reviewer’s comment. IPF is a complex, heterogeneous, and progressive disease of unknown etiology. Animal models don’t fully recapitulate physiologic findings of IPF or histopathologic pattern of usual interstitial pneumonia. However, they do enable mechanistic investigations relevant to fibrogenesis. According to an official American Thoracic Society workshop report, murine intratracheal bleomycin model is the best-characterized animal model available for preclinical experiments of pulmonary fibrosis [1]. IPF is a disease of advanced age, studies assessing bleomycin in older mice revealed more exuberant fibrosis, but this remained associated with enhanced inflammatory responses [2-3]. There is currently no evidence that the pathways leading to fibrosis are different in young compared with old mice, although older mice may resolve injury more slowly. Furthermore, histological changes associated with aging, such as increased airway thickness, may complicate histological assessments necessary to confirm biochemical endpoints of matrix deposition (e.g., hydroxyproline assays) [4]. In summary, given the significant practical hurdles associated with generation of aged mice, (e.g., time and cost), the panel found no currently compelling reason to recommend either standard or prioritized use of aged mice for pharmacological testing.

[1] Jenkins RG, Moore BB, Chambers RC, et al. An Official American Thoracic Society Workshop Report: Use of Animal Models for the Preclinical Assessment of Potential Therapies for Pulmonary Fibrosis.[J]. Am J Respir Cell Mol Biol, 2017, 56(5):667-679.

[2] Redente EF, Jacobsen KM, Solomon JJ, et al. Age and sex dimorphisms contribute to the severity of bleomycin-induced lung injury and fibrosis. Am J Physiol Lung Cell Mol Physiol 2011;301:L510–L518.

[3] Stout-Delgado HW, Cho SJ, Chu SG, et al. Age-dependent susceptibility to pulmonary fibrosis is associated with NLRP3 inflammasome activation. Am J Respir Cell Mol Biol 2016;55:252–263.

[4] Matsuoka S, Uchiyama K, Shima H, et al. Bronchoarterial ratio and bronchial wall thickness on high-resolution CT in asymptomatic subjects: correlation with age and smoking. AJR Am J Roentgenol 2003;180:513–518.

Comment 3: Authors may explain the BLM dose in terms of units, which is the standard way of explanation.

Reply 3: We thank the reviewer’s suggestion. We have changed the dosage unit of BLM to U, please see line 73 and 75 of the manuscript.

Comment 4: Authors should explain the isolation protocol of PPF. Also, are the cells validated for the expression of markers? Please show all details in the MS (probably in supplements). This data is very important!

Reply 4: We thank the reviewer’s suggestion. We have supplemented the information about the isolation protocol of PPF, as shown in 2.2 section. In addition, α-SMA is a biomarker of pulmonary fibroblasts, our results revealed that PPF cells isolated from BLM-treated mice (day 14) have higher protein expression levels of α-SMA than PPF cells isolated from NaCl-treated mice, and RG could inhibit the expression of α-SMA in BLM-PPF cells and has no significant effect on the expression of α-SMA in NaCl-PPF cells (Fig. S3).

Fig.S3. RG inhibited the α-SMA expression level in BLM-PPF cells instead of NaCl-PPF cells. NaCl-PPF and BLM-PPF cells were incubated with RG (2μM, 4μM) for 24 hours to analyze the α-SMA expression levels by Western blot. Densitometric analysis were showed beside. β-tubulin was used as a loading control. Data are means ± Standard Error of Mean (SEM), *P<0.05, **P<0.01, ***P<0.001 (one-way ANOVA), NS: nonsignificant.

Comment 5: TGF-β1 experiments – What are the cell culture conditions maintained for this experiment? Like serum levels and how long TGF-β1 treatment? These are key information, and they are missing.

Reply 5: We thank the reviewer’s suggestion. We have added the information of TGF-β1 experiments, as shown in “2.3 cell culture” section or related figure legends.

Comment 6: Statistics – Use of t-test is totally inappropriate. For treatments with more than 2 groups, authors should be doing one-way ANOVA, followed by post-hoc tests. Further, SEM is fine for in vitro experiments, instead of SD. Authors are required to re-analyze all data for ANOVA and SEM.

Reply 6: We thank the reviewer’s suggestion. We have re-analyzed all data by using one-way ANOVA followed by post-hoc tests, and changed the SD for SEM in all figures.

Comment 7: As explained in (1), stand-alone effect of RG is required for in vivo and in vitro. Since authors have started the RG treatment on Day 8, a small animal experiment is do-able (for 7 days of treatment).

Reply 7: We thank the reviewer’s suggestion. In in vitro experiments, Mlg and NaCl-PPF cells were treated with RG (0, 0.125, 0.5, 1, 2, 4, 8, 16, 32, 64, 128, 256 and 512µM), and the results showed that 2µM and 4µM RG have no significant cytotoxicity on fibroblasts (Fig.S2 A-B). In addition, RG did not inhibit the expression levels of α-SMA, Col1, p-Akt and p-Erk1/2 in Mlg and NaCl-PPF cells without TGF-β1 treatment (Fig.S1). In in vivo experiments, we have tested the animal toxicity of RG by multiple dose administration (15, 30, 150 and 300 mg/kg) once a day for 7 days, and the results showed that the weight and main organs (heart, liver, sleep, lung, kidney) of C57BL/6 mice had no significantly change (Fig.S2 C-E).

Comment 8: This is a must control experiment. In ideal condition, authors should have set up a dose dependent experiment first, choose a dose and then repeat the experiment with the chosen dose with and out RG for with and without BLM and go for a 2-way ANOVA.

Reply 8: We thank the reviewer’s comment. The results in our manuscript have indicated that RG (15mg/kg and 30mg/kg) could attenuate BLM-induced pulmonary fibrosis in mice and the high dose (30mg/kg) has better effect than low dose (15mg/kg). We have repeated the animal experiment with the chosen dose (30mg/kg) of RG with and without BLM and go for a 2-way ANOVA, and the data showed similar result with the dose dependent experiment (Fig.S4). In summary, RG could attenuate pulmonary fibrosis in animal model.

Fig.S4 RG attenuated BLM-induced pulmonary fibrosis in mice. (A) RG (30mg/kg) were given orally once a day from days 7-13 after BLM-treatment, lungs were harvested at day 14, and body weight loss was measured every day. (B) Percentages of fibrotic area in lung tissues. (C) Hydroxyproline contents in right lung tissues. (D) Forced vital capacity (FVC) in each group. Data in (A-D) are means ± Standard Error of Mean(SEM), *P<0.05, **P<0.01, ***P<0.001 (2-way ANOVA), NS: nonsignificant.

Comment 9: What is the effect of RG in fibroblasts (MIg and PPF) in the context of proliferation or cytotoxicity? This data is very critical. Secondly, what is the correlation between in vivo and invitro dose?

Reply 9: We thank reviewer’s comment. We determined the dosage of in vitro experiment by consulting the references and MTT experiments. We test proliferation or cytotoxicity of Mlg and NaCl-PPF cells by MTT assay in vitro, and our result revealed that RG inhibited fibroblasts proliferation in a dose-dependent manner and had not cytotoxicity in concentration of 2 and 4 µM (Fig.S2 A-B). In the reference [5], RG significantly decreased cell viability in a dose- and time-dependent manner (10, 20, 30, 40, 50µM) as compared to that noted in the control cells. In the reference [6], MTT assay showed that treated with RG at the concentration of 20μmol/L significantly inhibited gastric cancer cell growth as compared to control, and the inhibition effect showed dose dependent (5, 10, 20, 40µM). In the reference [7], RG did not show obvious cytotoxicity on the spheroids by 2.5 μg/ml (1.926µM) RG. Therefore, according to the above references and experiments, we selected the effective and non-toxic concentration 2μM and 4μM for in vitro studies of RG.

The in vivo and in vitro dosage of RG is relatively consistent. For example, hydroxyproline is an main content of tissue collage protein in mouse lung, and in vivo experiment showed that the inhibitory rate of RG in BLM-induced pulmonary fibrosis in mice at low and high dose are 73% and 65%. In vitro cellular result, Collagen type 1 is a biomarker of fibroblast activation, the inhibitory rate of RG in TGF-β1-induced Mlg cells activation at low and high dose are 85% and 48%. Therefore we considered that the dosage of in vitro is consistent with that of in vivo.

[5] Tsai J J , Pan P J , Hsu F T . Regorafenib induces extrinsic and intrinsic apoptosis through inhibition of ERK/NF-κB activation in hepatocellular carcinoma cells[J]. Oncology Reports, 2016.

[6] Lin X L , Xu Q , Tang L , et al. Regorafenib inhibited gastric cancer cells growth and invasion via CXCR4 activated Wnt pathway[J]. Plos One, 2017, 12(5).

[7] Mayer B , Karakhanova S , Bauer N , et al. A marginal anticancer effect of regorafenib on pancreatic carcinoma cells in vitro, ex vivo, and in vivo[J]. Naunyn Schmiedebergs Archives of Pharmacology, 2017.

Reviewer 3 Report

General comments:

The authors investigated the effects of regorafenib, a multi-kinase inhibitor, on bleomycin-induced pulmonary fibrosis and the mechanisms that mediate these effects. They found that regorafenib suppressed collagen accumulation and myofibroblast activation. Further in vitro mechanism studies showed that regorafenib inhibited the activation and migration of myofibroblast and extracellular matrix production mainly through suppressing the TGF-β1/Smad and non-Smad signaling pathway. The authors conclude that regorafenib attenuates bleomycin-induced pulmonary fibrosis via suppressing TGF-β1 signaling pathway.

General concerns:

  1. Methods: Please describe the rational of regorafenib dosages in in vivo study.
  2. Materials and methods: 2.1. Animals: The dosing schedules were not consistent. The mice were orally exposed to nintedanib (Macklin, China), RG (Dalian Meilun Biotechnology, China) once a day for days 8–14 and sacrificed on day 15. Figure legends 1qand 6: (A) Regorafenib (15mg/kg, 30mg/kg) and Nintedanib (100mg/kg) were given orally once a day from day 7-13 after BLM-treatment and lungs were harvested at day 14. (A) The mice were orally treated with RG (15mg/kg, 30 mg/kg) from day 7 to day 13 after BLM administration. Please clarify these discrepancies.
  3. Figure 6A expresses similar findings as those in Figures 1A, 1B, and 1C. Please clarify and merge them.
  4. Figure legend 6 on page 11, lines 318: “---extracted to analyze the inhibitory effect of RG on fibroblast migration---”. However, the inhibitory effect of RG on fibroblast migration was not shown in this Figure.

Author Response

Reviewer #3: The authors investigated the effects of RG, a multi-kinase inhibitor, on bleomycin-induced pulmonary fibrosis and the mechanisms that mediate these effects. They found that RG suppressed collagen accumulation and myofibroblast activation. Further in vitro mechanism studies showed that RG inhibited the activation and migration of myofibroblast and extracellular matrix production mainly through suppressing the TGF-β1/Smad and non-Smad signaling pathway. The authors conclude that RG attenuates bleomycin-induced pulmonary fibrosis via suppressing TGF-β1 signaling pathway. General concerns:

Comment 1: Please describe the rational of RG dosages in in vivo study.

Replay 1: We thank reviewer’s suggestion. We selected the dosage of RG according to the references and the toxic assays of RG. In the references [1], RG inhibited the in vivo growth of CCA cells in two animal models at for 30mg/kg, and RG significantly inhibited both SK-Hep1/luc2 and Hep3B2.1-7 tumor growth as compared with vehicle group at 20 mg/kg [2]. In addition, RG inhibited MM tumor cells growth in vivo at 30mg/kg [3]. In the chronic toxic assays, RG has no toxic effects on mice at 15mg/kg, 30mg/kg, 150mg/kg and 300mg/kg(Fig. S2C-E). Therefore, according to the above references and experiments, we considered that the effective pharmacological concentration of RG is in the range of 10-30 mg/kg and this concentration range is not toxic to mice, so we selected two concentrations of 15 and 30 mg/kg for in vivo studies in pulmonary fibrosis mice model.

[1]Yeh C N , Chang Y C , Su Y , et al. Identification of MALT1 as both a prognostic factor and a potential therapeutic target of regorafenib in cholangiocarcinoma patients[J]. Oncotarget, 2017, 8(69):113444-113459.

[2]Weng M C , Wang M H , Tsai J J , et al. Regorafenib inhibits tumor progression through suppression of ERK/NF-κB activation in hepatocellular carcinoma bearing mice[J]. Bioscience Reports, 2018:BSR20171264.

[3] Breitkreutz I, Podar K, Figueroa-Vazquez V, et al. The orally available multikinase inhibitor regorafenib (BAY 73-4506) in multiple myeloma [J]. Ann Hematol, 2018, 97(5): 839-849.

Fig.S2 RG had not toxic effect on Mlg and NaCl-PPF cells and C57BL/6 mice.

(A-B) Mlg and NaCl-PPF cells were incubated with RG (0, 0.125, 0.5, 1, 2, 4, 8, 16, 32, 64, 128, 256 and 512µM) for 24 h, and MTT assays were used to analyze the cytotoxicity effect of RG on the cells. (C) RG (15mg/kg, 30mg/kg, 150mg/kg and 300mg/kg) were given orally once a day for 7 days in normal mice, and the weight changes were shown from day 1 to day 7. (D) The appearance of mouse viscera. (E) The appearance of five main organs (heart, liver, spleen, lung, kidney).

Comment 2: Materials and methods: 2.1. Animals: The dosing schedules were not consistent. The mice were orally exposed to Nintedanib (Macklin, China), RG (Dalian Meilun Biotechnology, China) once a day for days 8–14 and sacrificed on day 15. Figure legends 1 and 6: (A) RG (15mg/kg, 30mg/kg) and Nintedanib (100mg/kg) were given orally once a day from day 7-13 after BLM-treatment and lungs were harvested at day 14. (A) The mice were orally treated with RG (15mg/kg, 30 mg/kg) from day 7 to day 13 after BLM administration. Please clarify these discrepancies.

Replay 2: We thank the reviewer to identify this mistake. The text of “Materials and methods: 2.1. Animals:----once a day for days 8–14 and sacrificed on day 15---” was fault due to our inattention, and we have reversed this sentence to “Materials and methods: 2.1. Animals:----once a day for days7–13 and sacrificed on day 14---”.

Comment 3: Figure 6A expresses similar findings as those in Figures 1A, 1B, and 1C. Please clarify and merge them.

Replay 3: We thank the reviewer’s suggestion. We have combined Figure 6A to Figure1.

Comment 4: Figure legend 6 on page 11, lines 318: “---extracted to analyze the inhibitory effect of RG on fibroblast migration---”. However, the inhibitory effect of RG on fibroblast migration was not shown in this Figure.

Replay 4: We thank the reviewer to identify this mistake. We have removed the inappropriate description.

Round 2

Reviewer 1 Report

The revised manuscript has been significantly improved, and concerns have been adequately addressed.

Reviewer 2 Report

No more suggestions